# A Relatively Simple Look at the Rather Complex Crystallization Kinetics of PLLA

**DOI:** 10.3390/polym15081880

**Published:** 2023-04-14

**Authors:** Jorge López-Beceiro, Ana-María Díaz-Díaz, Enrique Fernández-Pérez, Ignatius Ferreira, Walter W. Focke, Ramón Artiaga

**Affiliations:** 1Centro de Investigación en Tecnoloxías Navais e Industriais (CITENI), Campus Industrial de Ferrol, Universidade da Coruña, 15403 Ferrol, Spain; 2Department of Chemical Engineering, University of Pretoria, Pretoria 0002, South Africa

**Keywords:** PLA, crystallization, DSC, kinetics

## Abstract

This work demonstrates that, despite the existence of a significant number of works on PLA crystallization, there is still a relatively simple way, different from those already described, in which its complex kinetics can be observed. The X-ray diffraction (XRD) results presented here confirm that the PLLA under study crystallizes mostly in the α and α′ forms. An interesting observation is that at any temperature in the studied range of the patterns, the X-ray reflections stabilize with a given shape and at a given angle, different for each temperature. That means that both α and α′ forms coexist and are stable at the same temperatures so that the shape of each pattern results from both structures. However, the patterns obtained at each temperature are different because the predominance of one crystal form over the other depends on temperature. Thus, a two-component kinetic model is proposed to account for both crystal forms. The method involves the deconvolution of the exothermic DSC peaks using two logistic derivative functions. The existence of the rigid amorphous fraction (RAF) in addition to the two crystal forms increases the complexity of the whole crystallization process. However, the results presented here show that a two-component kinetic model can reproduce the overall crystallization process fairly well over a broad range of temperatures. The method used here for PLLA may be useful for describing the isothermal crystallization processes of other polymers.

## 1. Introduction

Poly(lactic acid) (PLA) is a thermoplastic polyester that is generally obtained from raw materials of plant origin. It is well known that PLA crystallization is a complex process that may involve various crystalline forms and transformations depending on the preceding thermal history and the thermal conditions under which the crystallization itself occurs [1,2,3,4,5,6]. From a crystallization perspective, PLA can be regarded as a copolymer of L- and D-lactic acid with the homopolymers being the extremes. Thus, the minor unit appears to act as a non-crystallizable comonomer causing the crystallization rate to decrease dramatically with increasing concentration of the minor unit [7]. The poly (L-lactic acid) (PLLA) homopolymer can crystallize in the α, β, and γ-forms, depending on the preparation conditions. The α-form of PLLA has a limiting disordered modification, defined as the α′-form. Racemic blends of PLLA and PDLA can form stereocomplex crystals, whose melting point is observed at about 50 °C above the melting point of homocrystals [8]. A nucleating effect of PLA stereocomplex on the crystallization of PLA homopolymers was observed [8,9,10,11]. Furthermore, the crystallization kinetics of PLLA is strongly dependent on the molecular mass, so that the crystallization rate drops strongly with increasing molecular mass [12]. In addition, the PLLA/PDLA stereocomplex has better mechanical properties than PLLA or PDLA homopolymers [1,2]. On the other hand, amorphous poly(D,L-lactic acid) (PDLLA) was investigated with the aim of producing biodegradable mosquito-repelling filaments with a reduced environmental impact [13].

PLLA crystallizes mostly in the α and α′ forms. Other two crystal forms, β and γ, were reported to develop under special processing conditions, whereas the α and α′ forms grow directly upon cooling from the melt. The two α and α′ polymorphs have very similar wide-angle X-ray diffraction (WAXD) profiles, the only differences arising from a small shift in peak position and from the appearance of very weak reflections. It was suggested that at Tc ≤ 95 °C PLLA crystallizes only in the α′ form; at 105 °C ≤ Tc ≤ 125 °C both α′ and α forms coexist; at Tc ≥ 145 °C only the α modification is present [3]. Considering that, in addition to the crystal structures, there is an amorphous region and a rigid amorphous fraction (RAF), the enthalpies of crystallization of the α and α′ crystals were calculated and found to be different. It was reported that the much weaker intermolecular bonding and higher free volume of the constrained RAF regions with respect to the crystal phase can favor chain rearrangement under some circumstances [14]. A lower value of the enthalpy of crystallization/melting of α′ form was observed, which seems reasonable due to the presence of conformational defects in the disordered α′ crystals. [15]. A number of factors including the structure of the melt, and the presence of self-seed remnants of former crystals were reported to affect the recrystallization rate [4]. Faster kinetics of the α′/α-transition were observed in the case of shorter macromolecules. That behavior was ascribed to a faster melting of smaller α′ crystals, faster growth of α crystals from the non-isotropic melt containing remnants/self-seed from molten α′ crystals, and/or a higher number of such α′ crystal remnants/self-seed [16]. Moreover, the melt temperature preceding the crystallization seems to have an important effect on the rate of recrystallization of α′-crystals so that the recrystallization rate decreases with increasing melt temperature due to the lowering of the homonuclei concentration. Particularly, it was found that homonuclei of the highest stability survive up to about 170 °C [4].

Two-dimensional WAXD and wide-angle neutron diffraction (WAND) were used to analyze the crystal structure of the PLLA α form [17]. Time-resolved simultaneous small-angle X-ray scattering (SAXS) and WAXD were used to track the unit cell parameters and the apparent crystal sizes in isothermal crystallization [18].

It has been confirmed that the small exothermic peak in the DSC curve detected, on heating, just prior to the melting peak, is associated with a first-order-type disorder-to-order (R′-to-R) phase transition [19]. The remarkable change in the half-width of the X-ray reflections corresponds to the change in the domain size between the R′ and R phase regions [19]. It has been proposed that the transformation of flexible chains into rigid conformational ordered segments could be mediated by local order structures or topological constraints. Very interestingly, it was proposed that a universal local order parameter or descriptor of polymers would be required to explain their crystallization behavior [20]. However, despite the numerous models and theories proposed so far for polymer crystallization, there is still no theoretical approach that can be considered generally satisfactory.

In addition, in the case of PLLA/PDLA racemic blends, formations of stereocomplex (SC) and homocrystals (HCs) are competing in the crystallization process of PLLA/PDLA racemic blend, when the crystallization temperature (Tc) is lower than the Tm of HCs [21].

This work aims to explain the isothermal crystallization kinetics of a PLA by a mixture function of the number of components that can be observed through the crystallization process. For that, crystal structures were examined by XRD in the experimental range of temperatures used for the isothermal crystallization. The crystallization data to be used for kinetic analysis were obtained from DSC experiments. Instead of XRD, the DSC heat flow signal was preferred for kinetics because it is sensitive to the reaction rate and can be continuously measured. Thus, the overall crystallization was tracked through the DSC curve and each DSC curve was fitted by the number of components corresponding to the number of crystals present in the sample.

## 2. Experimental Procedure

A commercial semi-crystalline fiber-grade poly(L-lactic acid) was used in the extrusion-compounding experiments. The Institute of Polymer Research in Dresden, Germany assisted with the characterization of this polymer sample. The molecular mass was determined with size exclusion chromatography (SEC) using chloroform as the mobile phase. The weight-average molar mass was 130 kDa and the number-average molar mass was (Mn) 44 kDa corresponding to a polydispersity of 2.95. The D-unit content was determined as 2.3% by polarimetry. The optical rotation of a 1.00 g dL^−1^ solution of the polymer was measured at a wavelength of 589 nm. 

### 2.1. DSC

All DSC experiments were carried out in a TA Instruments MDSC Q2000 furnished with a refrigerated cooling system. An N2 purge was applied throughout the experiments. Self-nucleation experiments were performed to find a temperature at which all the self-nuclei will be erased. A sample of about 10 mg was crimped in aluminum crucibles. The experimental procedure, based on the literature, consisted of heating the sample up to a temperature above the DSC melting temperature peak at which one tries to verify whether or not the self-nuclei have been destroyed [22,23,24,25]. Then the temperature is kept constant for two minutes and finally, the sample is cooled at 10 °C/min down to 20 °C, a temperature low enough to allow the sample to crystallize until saturation during the cooling ramp. The procedure was repeated for each of the temperatures at which such verification is sought. These temperatures were 205, 200, 195, and 190 °C. Once that temperature was determined to be 200 °C, it was chosen for the initial step of the subsequent isothermal crystallization experiments. Another sample of about 11 mg was crimped in aluminum crucibles for the isothermal crystallization experiments. This sample size was chosen to be similar to that of the self-nucleation experiments so that the crystal nuclei deletion proof is still valid for the isothermal crystallization tests. Thus, the sample was heated up to 200 °C in order to destroy any self-nuclei remnants that could survive at lower temperatures [4]. In fact, as commented in the experimental results section, at 195 °C there are still traces of self-nuclei. The procedure consisted of heating up to 200 °C at 5 °C/min. The sample was kept at 200 °C for 2 min and then cooled at 30 °C/min to the first crystallization temperature at which it was allowed to remain for enough time to complete most of the crystallization exotherm. The 30 °C/min cooling rate is relatively fast, which to some extent prevents crystallization from occurring before the isotherm. At the same time, by not being excessively fast, the chance of a thermal gradient and inhomogeneity is reduced. This cycle was repeated for all crystallization temperatures. Crystallization temperatures were chosen every two degrees in the range from 88 to 124 °C and the order of the isotherms was randomized. Two of the initial crystallization steps were repeated at the end of the series to check that the sample behavior was not altered as a consequence of the repeated heating–cooling cycles. In addition, to verify that the chosen sample size does not cause a significant thermal gradient in the sample, a crystallization experiment was conducted at 100 °C with samples of 11.70 and 6.42 mg.

### 2.2. XRD

The experiments were carried out in a Bruker Siemens D5000 diffractometer with Bragg–Brentano geometry and θ/20 configuration, equipped with a graphite monochromator. The instrument was provided with a homemade device to control the temperature of the sample in the XRD chamber. It consisted of a heater located under the XRD crucible, connected to an external controller to which a thermal camera was also connected, as illustrated in Figure A1 of the Appendix A. The thermal camera was placed above the sample and focused on it from a sufficient distance so as not to interfere with the operation of the XRD equipment. The system allows sample temperature and time data to be recorded on the same computer as the XRD data so that the temperature and diffraction records can be merged.

As in the case of the DSC experiments, the samples were heated up to a high enough temperature to destroy the remnants of homonuclei. The procedure consisted of heating the XRD sample crucible to the crystallization temperature through the electrical heater device. Then, the temperature control was deactivated and the sample is heated manually using a heat gun and carefully observing the temperature measured by the thermal camera. When the temperature exceeded 220 °C throughout the sample, the manual heating was suspended and the system control was re-activated. It deserves mention that most of the XRD bronze crucible, above which the polymer sample was placed, is at a lower temperature than the sample because the heat gun was mostly focused on the sample. The cooling to the crystallization temperature took about 5 min. Once the preset crystallization temperature was reached, diffractograms were obtained every 6 min. The temperatures chosen for isothermal crystallization were 90, 102, 116, and 124 °C. Once the patterns were stabilized at the isothermal temperature, the sample was allowed to cool to near room temperature.

## 3. Experimental Results

### 3.1. DSC

Figure 1 shows how heat flow exothermal peak shape changes from 190 to 200 °C, which is an indication that different amounts of self-nuclei exist in the different samples. At 195 °C there are still some remaining self nuclei. However, it can be seen that there is no difference between the plots obtained at 200 °C and 205 °C. This proves that at 200 °C there remain no more self nuclei. Consequently, that temperature was chosen for the initial treatment in the isothermal crystallization experiments. On the other hand, additional crystallization experiments performed with 6.42 and 11.70 mg samples showed no significant differences, as can be seen in Figure A2 of the Appendix A. This confirms that 11 mg is not an excessive size in this context.

Figure 2a shows the DSC isothermal plots obtained from the melt at the indicated temperatures. Figure 2b represents the heating scans just after the isothermal treatments represented in Figure 2a. The isothermal plots show two crystallization events, which follow different trends with temperature. A few crystallization and melting events are observed on the temperature scan plots; their locations on the temperature axis are strongly dependent on the temperature at which the sample was allowed to isothermally crystallize. It is remarkable that the variations in size and position of the peaks from one isothermal curve to another are continuous as the temperature changes, and there is no sudden appearance or disappearance of events with any moderate change in temperature. Something similar applies to the temperature scans as the size and position of the events on the time axis change smoothly from one to another of the curves obtained from isothermal treatments at near temperatures. These observations would be consistent with the co-existence of two crystal forms so that changes would result from the small changes of the predominance of one crystal form over the other as temperature changes. The melting peaks were observed in the 160 to 180 °C range, which is in line with other studies on PLLA [15]. That range of temperatures also corresponds to the melting of PLLA and PDLA homocrystals [21]. The ramps of low-temperature samples show clear cold crystallization exotherms and intermediate-temperature samples show double melting peaks. However, these ramp tests on samples subjected to different isothermal crystallization do not allow us to conclude by themselves which were the crystalline structures present at the end of each isothermal crystallization.

### 3.2. XRD

DSC analysis indicates that two components are needed to accurately fit the crystallization exotherms in the high-temperature range, from 114 to 124 °C. That is congruent with the fact that two crystal structures, α and α′, are formed in that range of temperatures, as reported in a number of works [3,12,14,19,26,27]. At lower temperatures, good fits are obtained with only one logistic component. It does not mean that only one crystal structure exists, but the contribution to the exotherm of one of them is practically negligible compared to the other. Thus, considering other reports, the predominant crystal structure in the low range of temperatures covered in this work is α′ [3]. In Figure 2, the green and yellow areas are shaded to highlight the apparent contribution of the α and α′ structures to the exothermic DSC peak. The α effect starts very weakly around 108 °C and becomes more evident with increasing temperature. However, the temperature increase also causes the two crystallization processes corresponding to α and α′ to appear more and more strongly overlapped. This overlap appears more marked at higher temperatures, which means that at these temperatures the two crystalline forms form simultaneously.

The XRD results here obtained show that, in general, the advancement of the crystallization process can be observed both on the 16.4° and the 18.6° diffraction peaks, the changes are bigger in the 16.4° one, as can be observed in Figure 3. It can be observed that the band of about 16.4 is growing with time and then stabilizes.

Figure 4 shows the stable diffractograms obtained in crystallization from the melt at the indicated temperatures. The peaks of the patterns at 89 and 102 °C are located on the left side with respect to those of the 116 and 124 °C patterns. That points to two different crystalline structures. The correspondence of XRD diffractograms to specific crystal structures was covered in a number of works [3,8,17,18,19,28,29].

## 4. Fittings and Data Analysis

Since only two crystal structures were identified by XRD in the temperature range covered in the isothermal crystallization DSC experiments, a two-component mixture model was adopted to fit the crystallization DSC curves. As described in previous works [6,30,31], for each isothermal component a time derivative of a generalized logistic (TDGL) accounts for the exotherm whereas a generalized logistic (GL) may account for a step change of baseline due to the heat capacity decrement inherent to the crystallization process. However, that baseline step change is generally very small compared to exotherm height and can generally be disregarded. In the DSC experiments shown in Figure 2a, since a cooling rate of only 30 °C/min was used, it is not possible to ensure that the crystallization process did not start during the cooling process preceding the crystallization isotherm. Moreover, it is well known that DSC signals are not very reliable in the vicinity of a change from linear heating to isotherm. Therefore, in practice, a reliable baseline preceding crystallization is not available. In these circumstances, we consider that it is preferable to assume a flat horizontal baseline that will be represented by a constant. Thus, each exothermic peak of isothermal crystallization will be represented by a TDGL
(1)$$y\left(t\right)=\frac{c\cdot b\cdot \exp\left(-b\cdot \left(t_{apm}-t\right)\right)}{{\left(1+\tau \cdot \exp\left(-b\cdot \left(t_{apm}-t\right)\right)\right)}^{\left(1+\tau \right)/\tau }}$$
where *t_apm_* represents the time at the exothermic peak maximum, *c* is the area of the peak, *b* is the rate factor, which depends on the temperature, and *τ* is the symmetry factor, with *τ* = 1 for perfect symmetry. That expression can be written as a function of conversion, *x*, and time
(2)$$y_{2}\left(t,\;\alpha \right){=c}_{t}\cdot b\cdot \exp\left(-b\cdot \left(t_{apm}-t\right)\right)\cdot \left(1-x\right)1+\tau$$

In the spirit of finding the simplest solution, we started by trying to fit each curve by a single Equation (1) TDLG function, as shown in Figure 5. The fittings were based on minimizing the average squared error (ASE) through the non-linear package of the Gnumeric software [32]. In order to include the temperature dependence of the rate factor, based on our previous work, a Gaussian distribution around a central temperature, *T_cent_*, was imposed [6]:(3)$$b\left(T\right)=\frac{1}{t_{cryst}}\cdot \exp\left(-ln\left(2\right)\cdot {\left(\frac{T-T_{cent}}{T_{hwhm}}\right)}^{2}\right)$$
where *t_cryst_* represents a characteristic crystallization time, which is the inverse of the maximum crystallization rate that is possible at any temperature; *T_cent_* is the temperature at which the maximum rate is obtained, and *T_hwhm_* is the half-width at height maximum, which is related to how the crystallization rate slows down as the crystallization temperature moves away from *T_cent_*.

An additional restriction consisted of constraining the symmetry parameter to take the same value at any temperature as long as it was the same crystallization process. This way, it was possible to fit all curves reasonably well in the 90 °C to 112 °C range. However, at higher temperatures there is an important shoulder at the beginning of the isotherm which makes it impossible to properly fit the entire curve by a single TDGL function. Thus, a new TDGL function was added to the model as displayed in Figure 6. Although Equations (1) and (2) are equivalent, the fit was performed with Equation (1) since the fitting is simpler if the conversion, *x*, is not required. For those exothermic peaks where a good fit with a single component is not evident, single- and dual-component fittings were performed and the resulting fits were compared both visually and by ASE. Whenever the quality of fit achieved with one function was similar to that achieved with two functions, the single-function fit was chosen. In the case of two components, two identical expressions of Equation (1) were used and the parameter values for each component were optimized. Before performing the optimization process, two constraints were imposed: First, for simplicity, and taking into account the shape of the exothermic peaks observed in the DSC curves, it is assumed that the left–right asymmetry of each crystallization process does not change significantly with temperature. This means that a single value of *τ* will be optimized for each component at all temperatures. Second, the parameter *b* in Equation (1) depends on temperature, as described in Equation (3). Therefore, taking into account these constraints, all isothermal curves were fitted simultaneously by the Gnumeric software using one or two functions described by Equation (1), where the parameter *b* was substituted by Equation (3). In the end, for each component unique values of *τ*, *t_cryst_*, *T_cent,_* and *T_hwhm_* were obtained for all temperatures. This means that the values of *b* are different at each temperature, following the trend described by Equation (3) with the optimized parameters. Likewise, *t_apm_* takes a different value at each temperature. Table 1 shows which TDGL functions, f1 or f2, were used for each temperature and the ASE values obtained in each case. According to the ASE values, similar quality of fittings was obtained in the low range of temperatures with only one function than in the high-temperature range with two components. Table 2 shows the parameter values, according to Equation (3), obtained for the rate parameter of each function from the simultaneous fitting of all curves. It is noticeable that although f1 reaches its maximum rate at 8 °C below f2, f2 is much faster, with a characteristic time a quarter of that of f1. However, although the b factor favors f2, for some other reason the structure represented by f2 does not actually form at temperatures below 110 °C. All fittings are presented in Figure A3 of the Appendix A. Figure 7 shows the dependence of the fitting parameters on temperature. As mentioned above, the c parameter represents the area of the crystallization peak for each mathematical component, f1 and f2. It is assumed here that each of these components corresponds to a different crystal form, α or α′. Thus, the c values represent the crystallization enthalpies of each crystal structure and should be proportional to the amount of each crystal structure formed. According to Figure 7a, the area of the crystallization process, represented by f1 + f2, generally increases with temperature. For the low-temperature range that total area is represented by a single function f1. That means that f1 represents α′, taking into account the present XRD results together with the literature reports mentioned above. Similarly, f2, which appears at 112 °C and its relative weight increases up to 124 °C whereas the weight of f1 decreases in the same range of temperature, can be easily related to the α structure. Figure 7b depicts how the rate factors change with temperature, according to the values in Table 2. In general, the values of *b* for f2 are much higher than those for f1. However, as commented in Figure 7a, regardless of the values of the rate factors, there is only the α′ component, f1, in the 90–110 °C range. At 112 °C the area of f2 was only 0.16, which means that the amount of α crystals formed at that temperature is practically negligible. Figure 7c plots the values of the *t_apm_* of f1 and f2. That time represents the time elapsed from the beginning of the isotherm to the peak maximum. These values resulted from the fittings of the curves and seem to follow parabolic trends, although there are no values of *t_apm_* in the 90–110 °C range because in that range of temperatures only one function was needed for the fittings. Similar trends were reported in other works for the half-time crystallization of PLLA as a function of the temperature of crystallization [33].

In addition, the symmetry parameter, *τ*, which was set constant for each process, resulted in very different values for each function, 0.23 for f1 and 7.86 for f2. Considering that a perfect symmetry corresponds to *τ* = 1, each process is skewed in the opposite direction with respect to the other, as can be observed in Figure 6. Considering Equation (2), *τ* can be theoretically related to a supposed reaction order. That would point to a high complexity of the crystallization process of the α structure, described by f2, with a value that is absolutely out of the range of values described for any type of reaction.

## 5. Conclusions

The XRD results presented here confirm that, as expected, the PLLA under study mostly crystallizes in the α and α′ forms when cooled from a high enough temperature to destroy the remnants of self-nuclei to a temperature in the 90 to 124 °C range. At any temperatures in that range, the X-ray diffractogram stabilizes with time but the reflections are shifted to slightly different angles depending on the crystallization temperature. That means that if only α and α′ forms are stable in that range of temperature, then the predominance of one over the other depends on temperature. Thus, a logistic-based two-component kinetic model was used to account for both α and α′ crystal forms.

The kinetics in the 90–112 °C range are explained with only one TDGL function, which represents the α′ structure, whereas two functions, representing both the α′ and α structures, are needed for the 114–124 °C range. The quality of the fittings was good in all cases. The symmetry parameter, which is related to a possible reaction order, suggests that the crystallization process of the α crystals is more complex than that of α′.

## Figures and Tables

**Figure 1 polymers-15-01880-f001:**
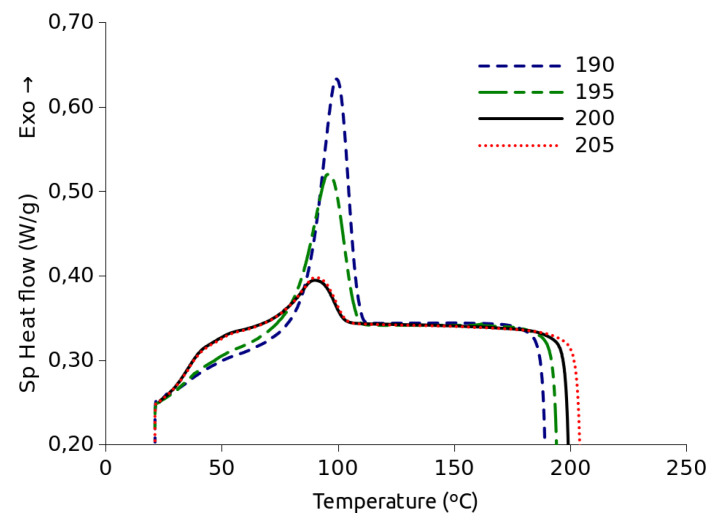
Plots of the DSC cooling scans from isothermal treatment at the indicated temperatures.

**Figure 2 polymers-15-01880-f002:**
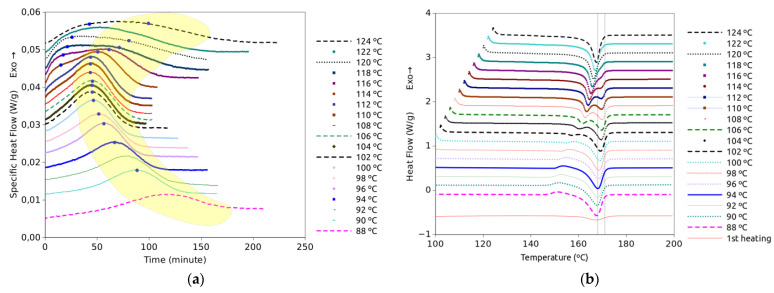
DSC isothermal plots obtained from the melt at the indicated temperatures (**a**) and plots from heating scans following each of the isothermal treatments (**b**). Vertical lines are aimed to better observe peak displacements.

**Figure 3 polymers-15-01880-f003:**
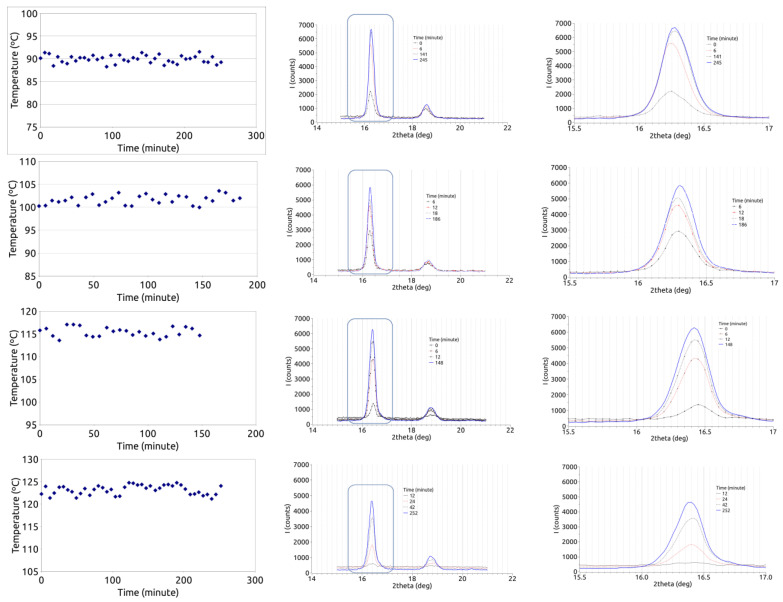
Temperature measured from the sample surface (**left**), sequence of diffractograms obtained at the temperatures shown in the left-hand graphs (**center**), and a zoomed area of the diffractograms (**right**).

**Figure 4 polymers-15-01880-f004:**
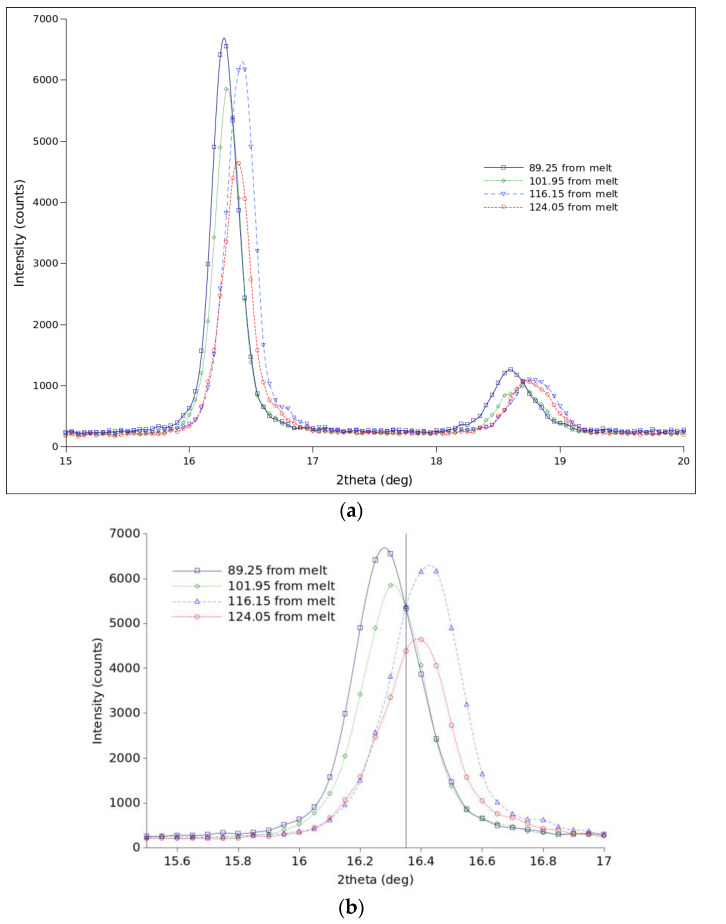
Diffractograms were obtained at the indicated temperatures (**a**). An enlarged area is displayed in (**b**).

**Figure 5 polymers-15-01880-f005:**
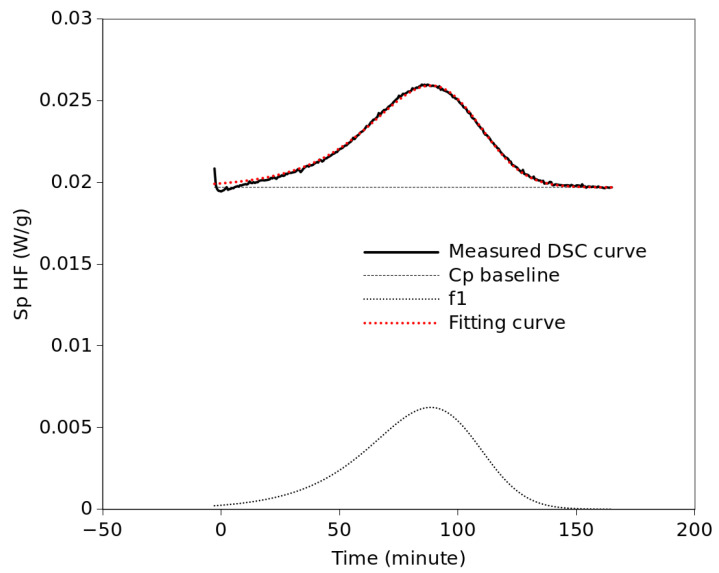
Fitting of a DSC curve by a single TDGL function.

**Figure 6 polymers-15-01880-f006:**
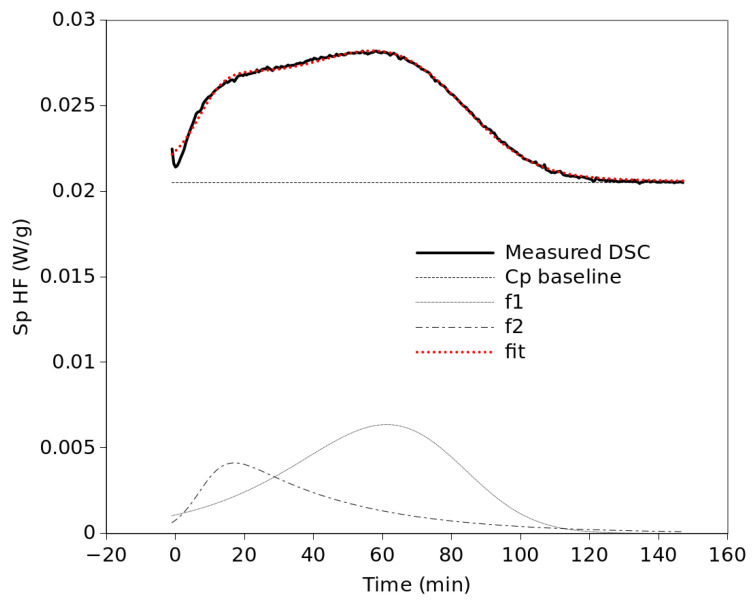
Fitting of a DSC curve by two TDGL functions.

**Figure 7 polymers-15-01880-f007:**
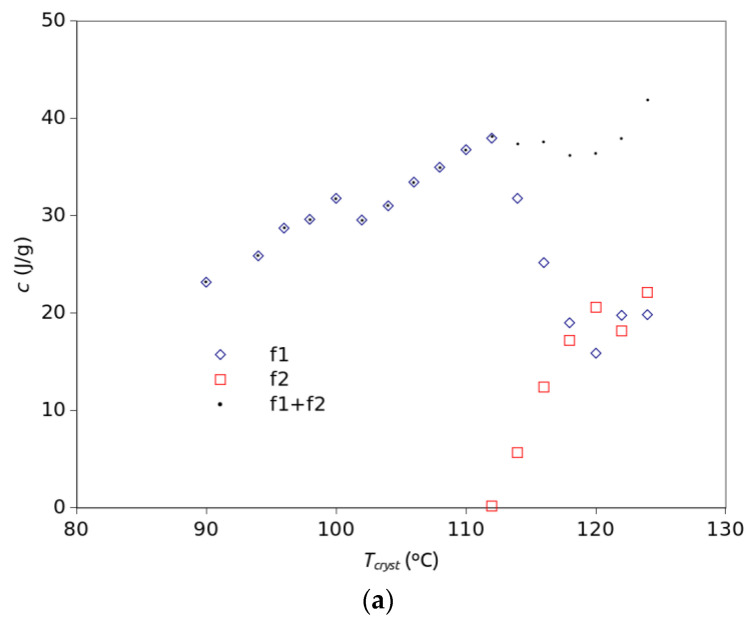
Temperature dependence of the fitting parameters: the *c* area factor (**a**), the *b* rate factor (**b**), and the *t_apm_* time at peak maximum, with over-imposed parabolic trend lines (**c**).

**Table 1 polymers-15-01880-t001:** Number of functions used for each temperature and the ASE values obtained in each fitting.

Temperature (°C)	TDGL Function Used	ASE × 1 × 10^8^
90	f1	1.51
94	f1	7.33
96	f1	3.01
98	f1	3.42
100	f1	6.90
102	f1	5.40
104	f1	6.85
106	f1	8.79
108	f1	1.26
110	f1	2.14
112	f1 and f2	4.58
114	f1 and f2	5.53
116	f1 and f2	2.33
118	f1 and f2	6.66
120	f1 and f2	6.66
122	f1 and f2	7.96
124	f1 and f2	5.15

**Table 2 polymers-15-01880-t002:** Parameter values, according to Equation (3), obtained for the rate parameter from the fittings of all curves.

	f1	f2
*t_cryst_* (s)	1016.95	229.90
*T_cent_* (K)	374.28	382.37
*T_hwhm_* (K)	21.42	12.00

## Data Availability

Data supporting reported results can be found at https://udcgal-my.sharepoint.com/:u:/g/personal/jorge_lopez_beceiro_udc_es/EaF-UFAFk0tAnKkdiVd1UX4BXVltl9YHisXwUGHduPwpaw?e=3yQeCa.

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
