# Peer review of "A Relatively Simple Look at the Rather Complex Crystallization Kinetics of PLLA"

_polymers, 2023, doi:10.3390/polym15081880_

Round 1
Reviewer 1 Report (New Reviewer)
Overall the work is interesting on studying the crystallization kinetics on PLLA through both DSC and XRD methods. There are some minor points and suggestions I have as follows:
- Please label the endo/exo direction with arrows for figure 1 and 2;
- The description of using electric heater for XRD measurement is not clear enough, can author provide a experimental set up figure at the supporting info. Or in the main context?
- The fitting of f1 and f2 with the TDGL method is not clear enough, how does that relate to equation 1 to 3? Please specify that.
Author Response
Please see attachment.

Reviewer 2 Report (Previous Reviewer 2)
If the significance is acceptable for the journal, I recommend its aceeptance.
Author Response
Thank you for recommending the article if the significance is acceptable for the journal.
This manuscript is a resubmission of an earlier submission. The following is a list of the peer review reports and author responses from that submission.
Round 1
Reviewer 1 Report
Reveiew of : A relatively simple look at the rather complex crystallization kinetics of PLLA
This work deals with the change in the crystalline structure of the PLA as a function of crystallization temperature. The authors investigated the effect of a wide range of crystallization temperatures Tc. Interesting results have been reported. The manuscript needs a meaningful number of corrections, modifications, and clarification to be accepted. The comments are bellow:
1. Page 1 lines 34-38, and following the context of PLA stereocomplex and crystallization in general (since the article deals with crystallization), I strongly recommend the authors to highlight the interesting nucleating effect of PLA stereocomplex on the crystallization of PLA homopolymers. Have look at this paper (https://www.mdpi.com/2073-4360/14/19/4092 ) and works from Prof. Pengju Pan (https://scholar.google.com/citations?user=unxxDtwAAAAJ&hl=en )
2. The authors are requested to mention the reference or the commercial name of the PLA grade under investigation. This is highly requested because the same PLA grade can be used and investigated in the future and researchers can know it and find it only through its commercial name.
3. The D-unit content is 2.3% which high enough to consider the grade under investigation as PLA and PLLA.
4. In the experimental procedure DSC part, the authors stated: “The samples were heated up to 200°C in order to destroy any self-nuclei remnants that can survive up to 170°C [4]”. Does the authors perform any self-nucleation experiments to define the borders of domain I (or the temperature at which all the self-nuclei will be erased). Because even if the reference 4 stated that PLA self-seeds can resist up to 170°C, this is not a general result that can be applied to all PLA grades. Some PLA grades have a melting point Tm above 170°C and for these PLA grades, the temperature at which all the self-nuclei will be erased is much higher than 170°C. The authors are requested to clarify this point and provided some experimental evidence about it (DSC results for example).
5. Always in the DSC experimental section, the authors stated: “This cycle was repeated for all the crystallization temperature”. The authors need to clarify whether they used only PLA sample for all the isothermal crystallization temperatures or not. The PLA is highly sensitive to degradation at the employed temperature (200°C), hence a clarification regarding the number of experiments performed for each DSC sample is needed.
6. Further details about the DSC instrument, DSC pans, and the employed DSC thermal protocol (such as the cooling rate) are requested.
7. Following comment 4, the DSC heating scans after isothermal crystallization showed melting points of around 170°C which confirms that the conclusion, about SN domains, obtained in reference 4 is not applicable in this study and using the current PLA grade.
8. There is no connection or continuity between the DSC results and the first part of the XRD results, especially regarding the defining of the nature of the crystal forms revealed by DSC and XRD. The authors are requested to rewrite the first part of the XRD results and make a connection with the DSC results. This is needed to enhance the overall quality of the paper.
9. Page 6 lines 187-189, the authors stated: “it is not possible to ensure … preceding the isothermal”. This is not true. In the case of PLA (a polymer with a low crystallization rate) and by using a proper cooling rate (between 60 and 70°C/min), the crystallization during cooling step can be suspended. The authors need to add further details about the thermal protocol in the experimental section and connect that with their DSC and XRD findings.
10. The writing of the equation seems blurry. Try to enhance it.
11. When talking about fitting and data analyses, there is a clear absence and luck of description of how the α and α’ changes with crystallization temperature. The authors are requested to correlate their fitting and data analyses with the changes in α and α’ content inside the sample.
12. During their explanation, the authors are requested to use the TERMINOLOGY α and α’ to express in a clear way the change in the crystalline structure by changing the crystallization temperature.
13. Reference 3 correct: Di Lorenzo M.L. instead of Lorenzo, M.L.D
14. Reference 4 correct: α′-crystals instead of A-crystals.
Author Response
The reply is attached

Reviewer 2 Report
This work deals with the crystallization kinetics of PLA in its a and a’ forms. There are already many research works appeared on the related issues. This makes the work less important and limited significance. Also the manuscript is poorly organized. Especially, the X-ray diffraction results are hardly reliable since the sharp diffraction peak reflect a very high crystallinity, which is not possible for the PLLA. Taken all these into account, I cannot recommend its acceptance.
Author Response
The reply is attached

Reviewer 3 Report
The reviewer does not see the novelty of this study and its suitability for publication in Polymers. I suggest the authors address these major concerns before submitting it to a suitable journal for consideration:
1- Please change “reflection” to “diffraction” in the XRD data and discussion.
2- The text and numbers in Figures 2 and 3 are too small, please change them accordingly for better vision. Please highlight the area zoomed in.
3- What is the RAF in this study? How do you compute/separate the contribution of RAF in crystalline domains and how does it contribute to the crystallization kinetics?
4- It is not clear what the kinetics of crystallization is in this study and what it means with the resulting fit data and numbers. Please add more details and discussion to explain the findings more deeply.
Author Response
The reply is attached

Round 2
Reviewer 1 Report
1) Regarding your choice of 200°C as a temperature to erase all the self-nuclei, yes, it is correct according to your DSC thermogram. On the other hand, the authors said “a temperature lower than 200 ºC does not destroy all self-nuclei remnants”. This is not not necessarily true. The conventional step between two different self-nucleation temperatures is only 2 degree and not 5 degrees. An interval of 5 degree is high enough to escape the exact temperature/boarder of the domain I of the SN. Hence, your choice (i.e., a temperature step of 05°C) is incorrect and the protocol you employed to define the boarder of the Domain I is wrong.
2) The use of a DSC sample with a mass of 11 mg for the isothermal crystallization study is incorrect. Polymers are known with their low thermal conductivity. And by using a sample of 11 mg (which is very big) and a cooling rate of 30°C/min (which is high), a high probability of a thermal gradient and non-homogeneity in the cooling will take place.
3) In page 5, the authors added the sentence “ That is congruent with….. in a number of works”. This statement is critical as well as not necessarily true and can not be written without any references. The authors are requested to highlight and reference the works they mentioned.
Reviewer 3 Report
The reviewer does not see its novelty as suitable for publication in Polymers.